# Our Clinical Experience in the Treatment of Myasthenia Gravis Acute Exacerbations with a Novel Nanomembrane-Based Therapeutic Plasma Exchange Technology

**DOI:** 10.3390/jcm11144021

**Published:** 2022-07-12

**Authors:** Dimitar Tonev, Radostina Georgieva, Evgeniy Vavrek

**Affiliations:** 1Department of Anesthesiology and Intensive Care, Medical University of Sofia, Academician Ivan Geshov Blvd 15, 1431 Sofia, Bulgaria; radostina.iv@abv.bg (R.G.); evavrek@hotmail.com (E.V.); 2Department of Anesthesiology and Intensive Care, University Hospital “Tsaritsa Yoanna-ISUL”, Belo More Str. 8, 1527 Sofia, Bulgaria; 3Neurological Intensive Care Unit, Department of Neurology, University Hospital “Tsaritsa Yoanna-ISUL”, Belo More Str. 8, 1527 Sofia, Bulgaria

**Keywords:** therapeutic plasma exchange, nanomembrane-based technology, myasthenia gravis, acute exacerbations

## Abstract

According to the American Academy of Neurology 2011 guidelines, there is insufficient evidence to support or refute the use of therapeutic plasma exchange (TPE) for myasthenia gravis (MG). The goal of this study was to determine whether a novel nanomembrane-based TPE could be useful in the treatment of MG. Thirty-six adult patients, MGFA 4/4B and 5, with acute MG episodes were enrolled into a single-center retrospective before-and-after study to compare a conventional treatment group (*n* = 24) with a nanomembrane-based TPE group (*n* = 12). TPE or intravenous immunoglobulins (IVIG) infusions were used in impending/manifested myasthenic crises, especially in patients at high-risk for prolonged invasive ventilation (IMV) and in those tolerating non-invasive ventilation (NIV). The clinical improvement was assessed using the Myasthenia Muscle Score (0–100), with ≥20 increase for responders. The primary outcome measures included the rates of implemented TPE, IVIG, and corticosteroids immunotherapies, NIV/IMV, early tracheotomy, MMS scores, extubation time, neuro-ICU/hospital LOS, complications, and mortality rates. The univariate analysis found that IMV was lower in the nanomembrane-based group (42%) compared to the conventional treatment group (83%) (*p* = 0.02). The multivariate analysis using binary logistic regression revealed TPE and NIV as independent predictors for short-term (≤7 days) respiratory support (*p* = 0.014 for TPE; *p* = 0.002 for NIV). The novel TPE technology moved our clinical practice towards proactive rather than protective treatment in reducing prolonged IMV during MG acute exacerbations.

## 1. Introduction

Myasthenia gravis (MG) is well known autoimmune disease in which antibodies bind to the postsynaptic acetylcholine receptors or related molecules in the neuromuscular junction, causing fluctuating muscle weakness [1,2,3,4,5,6]. The development of a stepwise approach to therapy and increasing use of immunosuppressive agents has led to increasingly good prognosis, quality of life, and survival in MG [6].

Myasthenic crisis (MC) is a severe presentation of MG in which patients experience a rapid deterioration of muscle control. Some authors use “myasthenic crisis” to refer to any exacerbation of MG which causes or threatens to cause frank respiratory failure [7]. Others use “myasthenic crisis” to refer solely to patients with MG exacerbation requiring respiratory support (intubation and mechanical ventilation or noninvasive positive pressure ventilation) [8,9]. The International Consensus Guidance for management of MG defines these as impending MC that could lead to crisis in days to weeks or manifest MC that represents worsening of myasthenic weakness requiring endotracheal intubation with invasive mechanical ventilation (ET-IMV) or noninvasive ventilation (NIV) to avoid intubation [6]. Regardless of the different interpretations, both are emergent situations requiring aggressive management and supportive care. Although generally accepted, these approaches require new evidence of their effectiveness, especially with the advent of innovative high-tech treatments.

The management of MC is challenging because of its fluctuant nature [9]. With improvement in respiratory care and intensive care unit management, the MC-associated mortality rate has declined from >40% in the early 1960s to approximately 5% today [10]. Immunologic therapies, including therapeutic plasma exchange (TPE), intravenous immunoglobulin (IVIG), and corticosteroids, are considered the mainstays of treatment during a MC. However, no consensus or standardized management for these patients has been established [9]. In addition, a small but significant proportion of MG patients remain refractory, lack tolerance, or develop side effects to steroids and immunosuppressants. Therefore, there is an unmet need for targeted immunomodulatory therapies, which has resulted in an ongoing campaign to develop safer and more effective treatments for MG [11]. The technological advances for a direct removal of auto-antibodies in patients with MG, such as immunoabsorption [12], double-filtration plasmapheresis [13], and nanomembrane-based TPE technology [14], pose new challenges and perspectives in this context.

Novel nanomembrane-based technology was approved by the American Society for Apheresis as a minimally invasive treatment. It is characterized by rapid control of quantitative and qualitative abnormalities of plasma and blood components using a semi-permeable nanomembrane that localizes immunologically active compounds on its surface [15]. According to the American Academy of Neurology 2011 plasmapheresis guidelines, there is insufficient evidence to support or refute the use of plasmapheresis for MG [7]. We reviewed our experience treating patients with MG exacerbations to assess the efficacy of TPE using a novel nanomembrane-based technology compared to the conventional treatment. The goal of this study was to determine whether a novel nanomembrane-based TPE could be useful in the treatment of MG.

## 2. Materials and Methods

We retrospectively identified all MG patients with impending or manifest MC admitted to an academic neuro-ICU between January 1999 and December 2019. The inclusion criteria were any acute exacerbations of muscle weakness leading to neuromuscular respiratory failure requiring noninvasive (NIV) or invasive (ET-IMV) respiratory support (or both), the presence of severe dysphagia with risk of aspiration, or of walking distance below five meters. Patients with congenital myasthenia, sepsis, renal failure, liver failure, malignancy, or incomplete data were excluded from the study.

Thirty-six consecutive MG patients with acute exacerbations, Myasthenia Gravis Foundation of America (MGFA) Class IV/IVB and Class V, aged ≥18, with 45 episodes of MG acute exacerbations were enrolled into a before-and-after single-center retrospective observational study. Two-thirds of them were treated with conventional treatment (conventional treatment group, *n* = 24), while the other third with a novel nanomembrane-based TPE technology (nanomembrane-based TPE group, *n* = 12). None of the patients were included in both groups in multiple admissions for MG exacerbations. The standard treatment comprised immunotherapies, including conventional TPE, IVIG, and corticosteroids, as well as symptomatic acute treatments as needed. The nanomembrane-based TPE group patients received the same standard treatment as their conventional treatment group counterparts, with the exception of using the novel nanomembrane-based TPE technology. This consisted of a “Hemophenix” apparatus using the ROSA nanomembrane (“Trackpore Technology”, Moscow, Russia). The nanomembrane was made of Lavsan film irradiated with accelerated charged argon particles. It has pores with 30–50 nm diameter and can eliminate molecules with weight less than 40 kDa. The device had an internal filling volume up to 70 mL, which can be used even in unstable hemodynamics, and the advantage of a single-needle access using any peripheral vein.

All patients were treated by a team of neurologists and anesthesiologists/intensivists in an academic neuro-ICU according to local guidelines [16]. TPE (3–5 procedures every other day) was performed as a rescue treatment in severe MG exacerbations at a high-dose regime in ventilator-dependent patients (removing 1–1.5 of the total plasma volume per treatment), or at a low-dose regime in spontaneously breathing patients (removing 0.3–0.5 of the total plasma volume per treatment). IVIG infusion was performed as a rescue treatment in less severe MG exacerbations in a dose regime of 2 g/kg over 2 to 5 days. The rescue therapies were used in impending or manifest MC, as well as selectively in patients at high risk of prolonged intubation and invasive mechanical ventilation (ET-IMV), along with anticholinergics, corticosteroids, and immunosuppressive drugs (according to the clinical scenario). The corticosteroid therapy included daily intravenous (i.v.) methylprednisolone at a low initial dose or at a high initial dose (according to the clinical scenario). In non-intubated patients, the treatment started with 20 mg i.v. methylprednisolone followed by 20 mg increments every 5–7 days, until there is marked clinical improvement or a dose of 100 mg per day was reached. In intubated patients, the treatment started with 100 mg i.v. methylprednisolone followed by 20 mg decrements every 10–14 days to a total dose 40 mg daily, when the patients switched to the equivalent oral dose methylprednisolone. The aggressive corticosteroid therapy was started in combination with fast-acting treatments (TPE and/or IVIG), as well as with azathioprine (in a few selected cases) [17]. An NIV trial was attempted in all patients with preserved swallowing needing respiratory support. Intubation was done after clinical and ABG considerations of acute respiratory failure or in the case of inability to maintain airway patency. An early tracheotomy (within 10 days, but predominantly at 24 to 48 h following the intubation) was performed in the event of expected prolonged MC as well. All tracheostomy cannulas were removed after the restoration of swallowing. All patients were subject to respiratory and general rehabilitation. The clinical improvement was assessed by manual chart review on forms filled in routinely upon admission and on transfer from the neuro-ICU using the Myasthenia Muscle Score (0–100), with at least a 20-point increase for responders [3].

Data regarding demographics, antibody status (anti-AchR, MuSK, seronegative), thymectomy, early-onset (<50 years), MGFA class before MC, Myasthenia Muscle Scores (MMS), comorbidities (Charlson Comorbidity Index, CCI > 2), recurrent MC, rates of intubations and early tracheotomies, non-invasive ventilation (NIV), invasive mechanical ventilation (IMV), time to successful extubation, neuro-ICU Length of Stay (LOS), hospital LOS, complications, and mortality rates were collected accordingly.

Primary outcome measures included: (1) changes in the implemented immunotherapies (TPE, IVIG, corticosteroids rates along with corticosteroid consumption); (2) changes in the implemented respiratory support rates; (3) changes in the implemented early tracheotomy rates; (4) changes in extubation time; (5) changes in MMS scores; (6) changes in neuro-ICU LOS; (7) changes in hospital LOS; (8) changes in the complication rate directly related to the implemented treatments; (9) changes in mortality rates.

Secondary outcome measures included: (1) distribution of potential variables of clinical relevance to the short-term (≤7 days) vs. long-term (≥8 days) respiratory support; (2) calculation of ORs of prognostic variables associated with short-term or long-term respiratory support.

Data were analyzed using SPSS v.20 (IBM SPSS, Armonk, NY, USA). Categorical data were compared with the χ^2^ test or Fisher’s exact test (as appropriate). Continuous variables were compared with the independent samples T-test in normal distribution or the Mann–Whitney U-test for distributions different from the normal (between-group comparisons), and with the paired samples T-test or Wilcoxon signed ranks Z-test, respectively (within-group comparisons). All statistical analyses were performed at ɑ = 0.05. To find independent predictors for short-term or long-term respiratory support, we dichotomized the MG patients into two groups (≤7 days vs. ≥8 days with implemented NIV/IMV or both). The distribution of potential variables of clinical relevance to the respiratory support were compared between the two groups using a univariate analysis. All the variables that achieved statistical significance were selected for the multivariate analysis. Thereafter, the *p*-values and OR values of the selected variables were calculated using binary logistic regression.

## 3. Results

Baseline demographics and clinical and laboratory characteristics of the studied groups were comparable (Table 1).

About 2/3 of the patients were women and 1/3 were men. One of the patients from the before-group had three exacerbations. Eight other patients from the before-group and six patients from the after-group had two exacerbations. The reasons for the worsening were anticholinesterase medication overdose (12.5% of the patients), other change in the treatment algorithm (8% of the patients), concomitant infections (45% of the patients), and surgery and stress (10% of the patients). The reason remained unknown in about 14% of the patients. In another 17% of them, the MC was the first manifestation of MG disease.

There were no between-group changes in survival and IVIG consumption, but there was a remarkable increase in the use of TPE (75%), less IMV, more aggressive NIV trials, and a reduction of early tracheotomy by one-half along with shortening the extubation time after the introduction of the novel TPE technology (Table 2). The only patient, who received conventional TPE in the before-group was intubated for two weeks, but had a good final outcome. Non-invasive ventilation was attempted in 37% in the before-group and in 58% in the after-group. The need for intubation and artificial ventilation was decreased dramatically in patients treated with the novel nanomembrane-based TPE (from 83% in the before-group to 42% in the after-group). In the after-group, the need for early tracheostomy was reduced to only 25%. The complications rate was similar—21% in the before-group vs. 25% in the after-group.

The mortality rate was not statistically significant—one patient (4.2%) in the before-group and one patient (8.3%) in the after group—both patients had significant comorbidities (Table 2). In the same way, age was revealed as a factor that adversely affected the responder rates in a time-dependent fashion. Patients with early-onset MG (<50 years) responded better to treatment than those with late-onset MG (>50 years) (Figure 1). The vast majority of early-onset MG patients were female (81%), while two-third of late-onset MG patients were male (66.6%). Both genders differ significantly between late-onset MG and early-onset MG counterparts (*p* = 0.004, same-sex comparisons).

Of 36 MG patients with acute exacerbations, 16 (44.4%) received short-term (≤7 days) respiratory support (including NIV, IMV, or both) (Table 3). By comparing baseline characteristics and implemented therapies between short-term and long-term respiratory support groups using univariate analysis, we found differences in patients’ age, MGFA class on neuro-ICU admission, and implemented TPE and NIV trials, respectively (*p* < 0.05). All variables that achieved statistical significance were further investigated using multivariate analysis.

The results of the binary logistic regression are shown in Table 4. The final equation suggested that lower MGFA class and implemented TPE and NIV trials were independent predictors of short-term respiratory support, whereas older age was the independent predictor associated with lower likelihood of short-term respiratory support.

No complications from the TPE itself were detected. We did not observe significant changes in blood, biochemical, or hemostaseologic parameters due to the plasmapheresis.

## 4. Discussion

Our study has some important caveats. Since we performed a single-center study with a relatively low number of patients, we have more preliminary findings than conclusive ones. Although our sample size was less representative, it corresponded to the observed small single-center study population sample size (13–53 patients) reported by others [5,9]. Another consequence of the single-center study is that some data are influenced by local peculiarities. In our previous work, we identified data that our myasthenic patients may search for help at a later stage of the disease. We have also found a high percentage of generalized MG in the Bulgarian population at the time of initial diagnosis [18]. The same conclusion was reached (two years later, independently from our previous work) by others as well. In a Japanese cohort, it was found that late-onset MG is predisposed to become generalized in the elderly [19]. Their study showed that elderly late-onset MG patients were more prone to severity (a finding supported by an observational Spanish study as well [20]), suggesting that these patients require aggressive immunomodulatory therapy, such as TPE or IVIG [19]. On the other hand, the International Consensus Guidance for management of MG suggests that TPE was favored (compared to IVIG) in patients with more severe respiratory impairment prior to initiating treatment [6]. Accordingly, TPE was more effective if initiated earlier after hospital admission in severe cases [21]. All these findings have a direct impact on our proactive treatment approach in terms of earlier implementation of nanomembrane-based TPE, especially in patients requiring ventilatory support due to the traditional opinion that the onset of action of TPE is more rapid than that of IVIG [22]. Rapid efficacy was especially important for our patients who were at risk of requiring intubation, or who had been intubated (because the risk of ventilator-associated complications increases with each additional day of invasive ventilation). However, our proactive nanomembrane-based TPE approach should be interpreted with caution and should be placed in the context of local peculiarities concerning our study population, local experience, availability, and insurance coverage.

Our primary outcomes (Table 2) support the advantages of TPE for superior ventilator status in terms of less intubation with IMV, more NIV, less tracheotomy, and quick response rate in MC reported by others [14,20,21,22,23,24,25]. Our secondary outcomes (Table 4) support the role of TPE for early extubation (≤7 days) as well as the role of NIV for decreasing ventilation duration (≤7 days) reported by others [9,26]. Our findings also suggest that older age, higher MGFA class on admission, and late-onset MG (Table 2, Figure 1) increase the risk of long-term ventilation, which is line with the findings reported by others [5,27]. Conversely, our results concerning predominantly male sex and late-onset MG non-responders (with response in the context of lower MMS scores) do not support the findings from another retrospective study, where the patients with male sex and late-onset MG were associated with a better response (with response in the context of complete resolution without need for maintenance TPE) [28]. The different results may be due to the selection bias concerning the baseline disease severity—we used TPE in MGFA class IV/IVB to V, whereas the opposite results were obtained when using plasmapheresis in MGFA class IIA to V.

Another aspect of the primary outcome measures included corticosteroid use as a part of our aggressive immunomodulatory approach (Table 2). A rapid-induction of high-dose corticosteroids is considered for MG patients with impending or manifested MC due to their faster onset of action in escalated doses [29]. In the neuro-ICU setting this approach is feasible because of the implemented NIV/IMV along with TPE/IVIG, which can counteract the temporary corticosteroid-worsening of myasthenic symptoms [30,31]. An interesting aspect of care in Japan is the use of high-dose 1.0 g/d i.v. methylprednisolone pulse therapy. The Japanese authors suggest that the combination of TPE with high-dose i.v. methylprednisolone could reduce the following corticosteroid oral doses as well [32]. Our relatively lower high-dose 100 mg/d i.v. methylprednisolone use along with the implemented novel TPE technology did not reduce the total (i.v. plus oral) corticosteroid consumption during the in-hospital treatment. However, we found a median dose of 30 mg methylprednisolone in our nanomembrane-based TPE group (equivalent to 38 mg prednisone) compared to a median dose of 620 mg methylprednisolone in the conventional treatment group (*p* = 0.109). Although not statistically significant, our finding could be clinically relevant and suggests the advantages of rapid improvement using more aggressive immunomodulatory therapies, leading to less high-dose corticosteroid exposure.

Our data suggest that the novel nanomembrane-based plasma exchange technology (NMBPE) is safe and efficient. It seems superior to the conventional treatment, especially in terms reducing the need for intubation, tracheostomy and the time on artificial ventilation, thus reducing the chance for the ventilator-dependent complications. The last factor was not detected in our study, probably because of the sample size limitation. The after-group was also characterized by increased use of non-invasive ventilation, thus suggesting more preserved swallowing in this group, which was not directly quantified in our study. The patients with NMBPE also had shorter courses of the disease.

One of the reasons that nanomembrane-based TPE is safer than the conventional TPE is simpler venous access (peripheral) and preserved hemodynamic stability of our patients. Of course, peripheral access is acceptable in some forms of conventional TPE as well [33].

Our data support the suggestion that nanomembrane-based TPE can improve outcomes. It can optimize rather than substitute intubation, ventilation, and early tracheostomy in severe MC [34]. However, our preliminary results suggest that TPE could focus our efforts on weaning patients off of mechanical ventilation rather than early tracheotomy. Aggressive management of myasthenia gravis (e.g., TPE) may enhance muscle strength and facilitate early extubation [27].

This pilot study leaves room for further questions. Some of them are connected with the most appropriate protocol for nanomembrane-based TPE. We have 3–5 procedure every other day, while for immunoabsorption and double-filtration plasmapheresis, some authors recommend a daily schedule scheme. They also suggest that the optimal number of sessions is four [35].

Another relevant question is whether the nanomembrane-based TPE would be also effective in other forms of treatments of MG, for example, in preparation for surgery. The clinical response of conventional TPE seems better in patients with higher MG scores [35], but more evidence is needed to confirm this finding with the novel mode of plasmapheresis as well. Likewise, further research is needed to confirm good efficacy in early-onset MG patients (<50 years) treated with double-filtration plasmapheresis [13] compared to early-onset MG patients (<50 years) treated with nanomembrane-based TPE (Figure 1). The nanomembrane-based TPE was successfully used in a myasthenic patient suffering acute respiratory distress syndrome (ARDS) after pneumonia as well [36]. New data in this direction would change the current consensus of the contraindication of therapeutic apheresis in MG patients with severe systemic infection [6].

An important research perspective could be to study the best combinations between nanomembrane-based TPE and other treatments. For example, the combination of double-filtration plasmapheresis and rituximab was proven to be effective in treatment of refractory MG [37], as were our combination of nanomembrane-based TPE and IVIG in this clinical scenario. The combination using double-filtration plasmapheresis proved to be more effective in lowering IgA levels than one using immunoabsorption [37]. The choice of these possible (but not exclusive) combinations could be influenced by the IgA levels. The IgA cut-off levels and other changes, such as the cellular immunity, need to be determined when applying nanomembrane-based TPE as well.

## 5. Conclusions

Nanomembrane-based TPE is a new and promising type of TPE, which seems to share the basic advantages of the conventional TPE, but probably with less adverse effects. The novel technology has changed our clinical practice towards more proactive rather than protective treatment in reducing prolonged IMV during MG acute exacerbations.

## Figures and Tables

**Figure 1 jcm-11-04021-f001:**
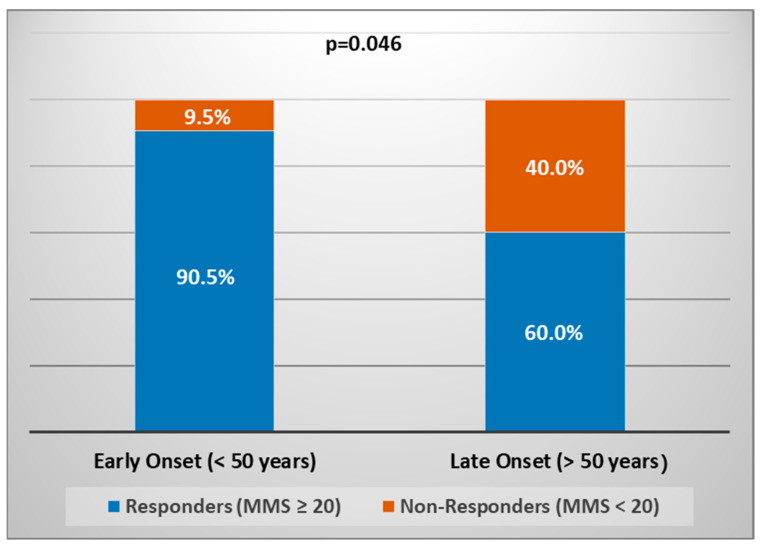
Proportions of responders and non-responders in early-onset (*n* = 21) and late-onset (*n* = 15) MG patients (MMS—Myasthenia Muscle Score).

**Table 1 jcm-11-04021-t001:** Demographics and clinical and laboratory characteristics (MGFA—Myasthenia Gravis Foundation of America; MC—myasthenic crisis; CCI—Charlson Comorbidity Index).

	Conventional Treatment *n* = 24	Nanomembrane-Based TPE *n* = 12	*p* Value
Gender (males/females)	8/16	6/6	0.471
Age (mean ± SD) (range)	41 ± 15 (18–76)	53 ± 17 (28–77)	0.078
Anti-AchR, *n* (%)	12 (50%)	7 (58%)	0.637
Anti-MuSK, *n* (%)	1 (4%)	2 (17%)	0.253
Double seronegative, *n* (%)	11 (46%)	3 (25%)	0.230
Thymectomy, *n* (%)	7 (29%)	5 (42%)	0.479
Early onset (<50 years), *n* (%)	15 (63%)	6 (50%)	0.473
MGFA class before MC (IV/V)	10/14	9/3	0.059
Cardiovascular disease, *n* (%)	5 (21%)	5 (42%)	0.192
Lung disease, *n* (%)	6 (25%)	3 (25%)	1.000
Kidney disease, *n* (%)	1 (4%)	2 (17%)	0.189
Diabetes mellitus, *n* (%)	1 (4%)	3 (25%)	0.061
Comorbidities (CCI > 2), *n* (%)	3 (12%)	5 (42%)	0.086
Recurrent MC, *n* (%)	9 (37%)	6 (50%)	0.473
Prior use of azathioprine, *n* (%)	7 (29%)	4 (33%)	0.808

**Table 2 jcm-11-04021-t002:** Treatment and outcomes before and after the introduction of the novel nanomembrane-based TPE technology (MMS—Myasthenia Muscle Score, ICU –intensive care unit, LOS—length of stay, VAP—ventilator-associated pneumonia, CPR—cardio-pulmonary resuscitation).

	Conventional Treatment *n* = 24	Nanomembrane-Based TPE *n* = 12	*p* Value
Therapy			
Escalated corticosteroids	12 (50%)	9 (75%)	0.282
Total dose corticosteroids [g (median)]	0.859 ± 0.959 (0.620)	0.235 ± 0.450 (0.030)	0.109
Intravenous immunoglobulin	6 (25%)	4 (33%)	0.700
Therapeutic plasma exchange	1 (4%)	9 (75%)	<0.0001
Non-invasive ventilation trial	9 (37%)	7 (58%)	0.236
Intubation with invasive ventilation	20 (83%)	5 (42%)	0.020
Early tracheotomy (≤10 days)	12 (50%)	3 (25%)	0.282
Outcomes			
Extubation time (days)	17 ± 21	5 ± 7	0.023
Responders (MMS ≥ 20)	18 (75%)	10 (83%)	0.691
Neuro-ICU LOS (days)	20 ± 24	10 ± 5	0.118
Hospital LOS (days)	28 ± 25	19 ± 11	0.470
Complications (VAP, atelectasis, CPR)	5 (21%)	3 (25%)	0.788
Mortality	1 (4.2%)	1 (8.3%)	0.618

**Table 3 jcm-11-04021-t003:** Potential factors of clinical relevance to short-term respiratory support during MG acute exacerbations (MGFA—Myasthenia Gravis Foundation of America; ICU—intensive care unit; CCI—Charlson Comorbidity Index).

	Short-Term Respiratory Support (≤7 Days) *n* = 16	Long-Term Respiratory Support (≥8 Days) *n* = 20	*p* Value
Baseline characteristics			
Gender (males/females)	6/10	8/12	0.878
Age (mean ± SD) (range)	53 ± 15 (32–76)	39 ± 15 (22–77)	0.010
Early onset (<50 years), *n* (%)	7 (44%)	14 (70%)	0.112
MGFA class on neuro-ICU admission (IV/V)	13/3	6/14	0.002
Comorbidities (CCI > 2), *n* (%)	4 (25%)	4 (20%)	1.000
Therapy			
Escalated corticosteroids	8 (50%)	13 (65%)	0.364
Intravenous immunoglobulin	3 (19%)	7 (35%)	0.456
Therapeutic plasma exchange	8 (50%)	2 (10%)	0.011
Non-invasive ventilation trial	12 (75%)	4 (20%)	0.001

**Table 4 jcm-11-04021-t004:** Predictors of short-term respiratory support during MG acute exacerbations (MGFA—Myasthenia Gravis Foundation of America; ICU—intensive care unit).

Predictors	OR	95% CI of OR	*p* Value
Age	0.942	0.896–0.990	0.018
MGFA class on neuro-ICU admission (IV/V)	10.111	2.086–48.999	0.004
Therapeutic plasma exchange	9.000	1.550–52.266	0.014
Non-invasive ventilation trial	12.000	2.484–57.975	0.002

## Data Availability

All data (accessed on 1 January 2020) are available under request to Dimitar Tonev (dgtsofia@abv.bg).

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
