# Peer review of "Our Clinical Experience in the Treatment of Myasthenia Gravis Acute Exacerbations with a Novel Nanomembrane-Based Therapeutic Plasma Exchange Technology"

_jcm, 2022, doi:10.3390/jcm11144021_

Round 1

Reviewer 1 Report

The paper improved after corrections.

No further comments.

Reviewer 2 Report

The authors compared the outcome of patients treated with conventional plasma exchange (PE) (24 patients) (they called this before group) versus nanomembrane based plasma exchange (12 patients) (they called this after group). Among patients presented with acute myasthenia exacerbation or crisis. They reported reduction in the need of invasive ventilation and earlier extubation in favor of nanomembrane PE. There was also trend toward less chance of early tracheostomy (but not statistically significant.).

The idea presented in this manuscript is interesting and worth to be published. However, the writing and the data presentation needs a lot of improvement. It is not presented well and not ready for publication at the current state. The sample size is not large so that should be compensated by more accurate and clear data presentation. Here is examples of things that need to be addressed:

11)      Abstract is not structured: no methods section and no results section.

22)      Abstract did not present the data in the appropriate statistical format. For example; they wrote:

“The univariate analysis found a remarkable 23 increase in the use of TPE (71%) and NIV trial (58%), and a reduction of early tracheotomy by one- half after the introduction of nanomembrane-based TPE. “

The traditional way of writing is that “ the need of invasive mechanical ventilation was less with after group (42%) compared to 83% in the before group, p value =0.02”

They reported reduction in tracheostomy by half . they were suppose to write down the number and by the way it was not statistically significant and hence this should be omitted from the abstract or at minimal to mention the p value according to table 2.

33)      The data presented in the abstract does not include the primary outcome mentioned in the body of the manuscript. Usually the primary outcome is mentioned in the abstract.

44)      The abstract included a lot of background information while to should include more methods and specific data of the results.

55)      The group naming: before group and after group is confusing. As it not representative of the intervention of interest. It should be for example conventional PE group and nanomembrane PE group.

66)        Need to specify if there were patients repeated between before and after group. Or they were totally different patients.

77)      One of the primary outcomes: rate of use of immunotherapies. Need to be more specific and object. For example: use of IVIg, Prednisone dose, … etc.

88)      Table -1 mentioned comorbidities: it need be mentioned separately: diabetes, hypertension, heart disease, lung diseases, … etc. instead of comorbidities.

99)      In table 2: escalated corticosteroid, what was the median of prednisone dose or what was the percentage on high prednisone dose > 40 mg for example. They did not define what is meant by escalated corticosteroid.

10)   Need to know the prior use of azathioprine or mycophenolate or methotrexate in each group

11)   Need the median prednisone dose in each group

12)   Line 196: “The final equation suggested that lower MGFA class, implemented TPE and NIV trial were independent predictors of short-term respiratory support, whereas older age was the only variable associated with lower likelihood of short-term respiratory support”.  The sentence needs clarification, why did not include age among the other 3 factors.

13)   Language issues examples:

a.       Line 202: “one complication from the TPE itself were detected.” It should be no complicatons.

b.       Line 213: “he same conclusion was reached (two years later, regardless of our previous work) by others as well. “ regardless of our previous work is not common phrase in such context.

c.       In line 226:’ because the risk of ventilator associated complications rises with each additional day”. Should be because the risk of ventilator associated complications inceases with each additional day.

In line 274: early onset < 40 years. It should be < 50 years

Author Response

This manuscript is a resubmission of an earlier submission. The following is a list of the peer review reports and author responses from that submission.

Round 1

Reviewer 1 Report

I think the subject of the paper is important and interesting. The introduction is too long and not focused. The description of the methods needs important improvement, it is not clear how the groups were chosen. The group size is very small and not equally distributed. And conclusions should be more cautious. 

Reviewer 2 Report

This study aims to unravel the clinical relevance of the use of plasmapheresis for MG. The authors used a before-after single-center retrospective observational study to compare the conventional treatment with advanced one using a novel nanomembrane-based TPE. Although the idea is interesting, the design and statistics need further consideration. There were some concerns as follows.

1. The Abstract and Introduction parts were not clear to highlight the importance of this study, including the logic and language.

2. The classification for the conventional treatment and advanced treatment is a little disordered, because too many subtypes of conventional treatments were included. This is not a religious group. In this case, the statistics were not convincing.

3. The observation index for the MC, such as the NICU length of stay, was not a direct evidence.

4. The conclusion is beyond the result implication, especially the superiority of “preventive” function. 
